# On the relation between apparent ion and total particle growth rates in the boreal forest and related chamber experiments

Loïc Gonzalez Carracedo[1], Katrianne Lehtipalo[2,3], Lauri R. Ahonen[2], Nina Sarnela[2], Sebastian Holm[2], Juha Kangasluoma[2], Markku Kulmala[2,4,5,6], Paul M. Winkler[1] and Dominik Stolzenburg[2]

[1]Faculty of Physics, University of Vienna, 1090 Vienna, Austria
[2]Institute for Atmospheric and Earth System Research/Physics, University of Helsinki, 00014 Helsinki, Finland
[3]Finnish Meteorological Institute, 00560 Helsinki, Finland
[4]Aerosol and Haze Laboratory, Beijing Advanced Innovation Center for Soft Matter Sciences and Engineering, Beijing University of Chemical Technology (BUCT), Beijing, China
[5]Joint International Research Laboratory of Atmospheric and Earth System Sciences, School of Atmospheric Sciences, Nanjing University, Nanjing, China
[6]Faculty of Geography, Lomonosov Moscow State University, Moscow, Russia

*Correspondence to*: Dominik Stolzenburg (dominik.stolzenburg@helsinki.fi)

**Abstract.** The understanding of new particle formation and growth processes is critical for evaluating the role of aerosols in climate change. One of the knowledge gaps is the ion-particle interaction during the early growth process, especially in the sub-3nm range, where direct observations are sparse. While molecular interactions would imply faster growth rates of ions compared to neutral particles, this phenomenon is not widely observed in the atmosphere. Here, we show field measurements in the boreal forest indicating a smaller apparent growth rate of the ion population compared to the total particles. We use aerosol dynamics simulations to demonstrate that this effect can be caused by a changing importance of ion-induced nucleation mechanisms during the day. We further compare these results with chamber experiments under similar conditions, where we demonstrate that this effect critically depends on the abundance of condensable vapors and the related strength of ion-induced nucleation. Our results imply that atmospheric ion growth rate measurements below 3 nm need to be evaluated very carefully as they do not represent condensational growth alone but are influenced by ion-particle population interactions.

## 1 Introduction

Cloud condensation nuclei (CCN) impact significantly the Earth's radiative balance by modifying the albedo of clouds (Twomey, 1974) and their mean lifetime in the atmosphere (Albrecht, 1989). New particle formation (NPF) by gas-to-particle conversion is frequently observed around the globe (Kulmala et al., 2004a; Kerminen et al., 2018, Lee et al., 2019) and contributes significantly to the total particle number concentration in the atmosphere (Merikanto et al., 2009; Spracklen et al., 2008). The new particles formed during NPF events have to grow fast enough to avoid coagulation loss with the larger pre-existing aerosols (Pierce and Adams, 2007; Kuang et al., 2009). The growth process is important to characterize because it determines the atmospheric significance of NPF events, with respect to the CCN budget (Gordon et al., 2017) and air

quality (Guo et al., 2014). The survival of growing atmospheric particles can be approximated by a competition between the growth rate and coagulation sink of these particles (see e.g. Kerminen & Kulmala, 2002). Therein, the growth rate is defined as the change of aerosol particle or ion diameter per time following Eq. (1):

$$\text{GR } [\text{nm } h^{-1}] = \frac{\Delta d_p}{\Delta t} \approx \frac{d_{p,f} - d_{p,i}}{t_f - t_i}, \tag{1}$$

where $d_p$ is the particle diameter in nm and $t$ the time associated with this particle diameter. However, this definition is not unambiguous. While the growth rate of a single aerosol particle can be theoretically calculated from vapor molecule condensation (e.g. Nieminen et al, 2010), atmospheric measurements do not track the growth of a single particle, but infer the growth rate from the change of the particle population over time in a large area (Kulmala et al., 2012). Different methods for the quantification of such an apparent particle growth rate exist (Dal Maso et al., 2005, Hirsikko et al., 2005; Lehtipalo et al., 2014), but direct comparisons of the different methods are sparse (e.g. Yli-Juuti et al., 2011). Moreover, due to the availability of ion size-distribution measurements below 10 nm with instruments like the neutral cluster and air ion spectrometer (NAIS; Manninen et al., 2009; Mirme and Mirme, 2013, Manninen et al., 2016), atmospheric growth rates have been often calculated in that size-range from the evolution of ion populations instead of the total (neutral plus charged) particle size-distribution (e.g. Manninen et al., 2010). However, the ion population and its time-evolution depend crucially on charging processes, ion-ion recombination and have an additional ion-induced nucleation source term (Leppä et al., 2009; Gonser et al., 2014). While charged particles are expected to grow faster by vapor condensation due to vapor-charged particle interactions which increase the collision cross-section (Leppä et al., 2011; Lehtipalo et al., 2016; Stolzenburg et al., 2020) this effect is typically not observed in the real atmosphere (Gonser et al., 2014; Kulmala et al., 2013a; Manninen et al., 2009).

Here, we use measurements of total particle and ion growth rates from the SMEAR II station in Hyytiälä, Finland and the CLOUD (Cosmics Leaving Outdoor Droplets) experiment at CERN (European Organization for Nuclear Research) to investigate the effect of ions on apparent nanoparticle growth rates. We additionally deploy aerosol dynamics simulations including ion processes (Leppä et al., 2009) to explain the observations and investigate the possible origin of the differing apparent ion and total particle growth rates for both settings.

## 2 Instrumental setup and theoretical approach

### 2.1 Field measurements and chamber experiments

Our field data was collected between March-September 2020 at the SMEAR II station based in the boreal forest in Hyytiälä, Finland (61°51' N, 24°17'E, 181 m a.s.l.), where we recorded 50 NPF events, of which for 18 we could quantify GRs in both the DMA-train and the NAIS from 1.8-8 nm (called "strong" NPF events in the following). The SMEAR II station is considered a semi-clean boreal forest environment because of the relatively long distance (>80 km) to major urban areas (Hari and Kulmala, 2005). The site is surrounded by a rather homogenous Scots pine forest and is equipped with

comprehensive instrumentation for measuring interactions between the forest ecosystem and the atmosphere. It is also part of the European Aerosols, Clouds and Trace gases Research Infrastructure (ACTRIS). The complete description of the Hyytiälä forest station site is presented in Hari & Kulmala (2005).

We additionally use data from the CERN CLOUD experiment, which allows precise control of the experimental conditions (relative humidity, temperature and trace gas concentrations). Furthermore, two electrode meshes in the chamber allow to establish neutral conditions by the application of an electric field which removes all air ions. A detailed description of the CLOUD experiment can be found in Duplissy et al., 2016. Here we use data from experiments which simulated the NPF process in Hyytiälä as close as possible (Lehtipalo et al., 2018), using a mixture of sulfuric acid, ammonia, NOx and oxidized organics from alpha-pinene and delta-3-carene ozonolysis as particle precursors (in total 14 experiments). The experimental conditions in Hyytiälä and CLOUD are compared in Table 1.

**Table 1 Overview of the variation of different parameters and condensable vapour concentrations important for nanoparticle growth in Hyytiälä (NPF-days only), CLOUD and assumed set values for the ion-UHMA simulations.**

| | Hyytiälä | CLOUD | ion-UHMA Hyytiälä-diur. | ion-UHMA Hyytiälä-param. | ion-UHMA CLOUD-param. |
|---|---|---|---|---|---|
| T [K] | 273-293 | 278 | 278 | 278 | 278 |
| rH [%] | 20-92 | 38 | 50 | 50 | 38 |
| $H_2SO_4$ [cm$^{-3}$] | $2\cdot10^5$-$9\cdot10^6$ | $1.2\cdot10^6$-$4.6\cdot10^7$ | $1\cdot10^5$-$4\cdot10^6$(**) | $1\cdot10^5$-$4\cdot10^6$(***) | $1\cdot10^5$-$1\cdot10^7$(***) |
| $NH_3$ [pptv] | 50-150(*) | 2-3000 | 150 | 150 | 100 |
| $HOM_{dim}$ [cm$^{-3}$] | $1.4\cdot10^4$-$1.1\cdot10^6$ | $3.4\cdot10^4$-$5.5\cdot10^6$ | not used | not used directly | not used directly |
| $HOM_{tot}$ [cm-3] | $6\cdot10^6$-$2.6\cdot10^8$ | $6.7\cdot10^5$-$4.3\cdot10^7$ | $2\cdot10^7$ | $2\cdot10^7$ | $2\cdot10^7$ |
| $[H_2SO_4]^2\cdot[NH3]\cdot[HOM_{dim}]$ | $2\cdot10^{17}$-$5\cdot10^{20}$ | $9.6\cdot10^{17}$-$1.1\cdot10^{24}$ | not used | $2\cdot10^{20}$-$2.5\cdot10^{23}$(***) | $2\cdot10^{19}$-$2\cdot10^{23}$(***) |
| $Q_{i.p.}$[i.p. cm$^{-3}$ s$^{-1}$] | 6-12 | 2-4 | 3 | 3 | 3 |
| $N_{i.p}$ [i.p. cm$^{-3}$] | 500 | 1000-2000 | 667-1437 | 621-1282 | 1594-1975 |
| J [cm$^{-3}$ s$^{-1}$] | 0.01-12.9 | 0.01-73.5 | 0-2 | 0-21.9 | 0-18.2 |
| | | | (Eq.(5)) | (**** Eqs. (2)&(4)) | (**** Eqs. (2)&(4)) |

(*) assumed from Makkonen et al. (2014), (**) predefined diurnal pattern, (***) analytical approximation to the available gas-phase measurements (****) analytical approximation to the parametrized nucleation rates

The usage of the CLOUD data enables a comparison of the effect of ions on particle growth under ambient and controlled laboratory conditions. Based on the chamber experiments, Lehtipalo et al. (2018) proposed parametrizations for particle formation (neutral formation rate at 1.7 nm, $J_{1.7}$(neutral)) and growth processes (growth rate GR) for the conditions similar to the boreal forest, which are given below:

$$J_{1.7}(\text{neutral}) \ [cm^{-3}s^{-1}] = a_1 \cdot [H_2SO_4]^2[NH_3][HOM_{dim}], \tag{2}$$

$$GR \ [nm \ h^{-1}] = b_1[H_2SO_4] + b_2[NH_3][H_2SO_4] + b_3[HOM_{dim}], \tag{3}$$

with the fitted constants $a_1 = 7.4 \cdot 10^{-23}$ s$^{-1}$pptv$^{-3}$cm$^6$ for the formation rate parametrization and $b_1 = 2.07 \cdot 10^{-7}$ nm h$^{-1}$cm$^3$, $b_2 = 7.3 \cdot 10^{-11}$ nm h$^{-1}$cm$^3$pptv$^{-1}$, $b_3 = 2.6 \cdot 10^{-6}$ nm h$^{-1}$ cm$^3$ for the growth in the size-range of

1.8-3.5 nm. In the above parametrizations, the sulfuric acid $[H_2SO_4]$ and highly oxygenated organic molecule (HOM) dimer $[HOM_{dim}]$ concentrations are given in cm$^{-3}$ and the ammonia mixing ratio $[NH_3]$ in pptv. We fit the ion-induced nucleation fraction by a function which is limited by the ion-pair production rate (as it gives the maximum rate at which ion-induced nucleation can proceed with every ion seeding a new particle) and which approaches the neutral nucleation rate exponentially around the vapor concentrations where ion-induced nucleation becomes less dominant:

$$J_{1.7}(\text{ion}) \ [cm^{-3}s^{-1}] = c_1 - c_1 \cdot \exp(c_2 \cdot [H_2SO_4]^2[NH_3][HOM_{dim}]) \tag{4}$$

We find $c_1 = 3.4$ cm$^{-3}$ s$^{-1}$ (close to the ion-pair production rate $Q_{i,p}$) and $c_2 = 2 \cdot 10^{-22}$ cm$^{-9}$ pptv$^{-1}$ (free parameter of the fit) using the $J(\text{tot}) = J(\text{ion}) + J(\text{neutral})$ data obtained under galactic cosmic ray conditions (no ion removal in the chamber) from Lehtipalo et al. (2018) for the fit.

## 2.2 Particle instruments

In both experimental settings we used a similar array of particle- and ion-size distribution measuring instrumentation.

### 2.2.1 DMA-Train

A DMA-Train is deployed to measure the particle size distribution between 1.8-8 nm. It contains six Grimm Aerosol GmbH S-Differential Mobility Analyzers (DMA) set to a fixed voltage to measure continuously at six different particle diameters between 1.8 and 8 nm (Stolzenburg et al., 2017). This configuration of the instrument allows a high temporal resolution and

a good sensitivity towards low particle concentrations in the sub-10 nm range. Furthermore, the DMA-train can measure also sub-3 nm particle growth with an unprecedented sizing precision due to the usage of mobility spectrometry. In Hyytiälä, the DMA-train was operated in a measurement container with a 1 m stainless steel inlet at a total inlet flow of 20 lpm to reduce sampling losses. Two TSI Model 3088 Soft X-Ray neutralizers were used to obtain the total (neutral plus charged) particle size-distribution from 1.8-8 nm. The DMA-train measurements from the CERN CLOUD chamber have been previously

reported in more detail (e.g. Stolzenburg et al., 2018, 2020), but the setup was overall very similar to Hyytiälä.

### 2.2.2 DMPS

The particle size distribution between 3 and 1000 nm was measured with a twin-Differential Mobility Particle Sizer (DMPS) in Hyytiälä. The twin-DMPS consists of a long and a short Vienna DMA and two butanol condensation particle counters (TSI 3025 and TSI 3775). The setup at SMEAR II is described by Aalto et al. (2001). The DMPS was located in a small

measurement hut ca. 20 m away from the DMA-train container and the DMPS inlet is inside the forest canopy on the roof of the hut at 8 m height. At CLOUD the total particle size-distribution above 8 nm was recorded with a TSI nano-SMPS (Tröstl et al., 2015) and a custom built long-SMPS.

### 2.2.3 NAIS

The ion size distribution was measured with an NAIS (Manninen et al., 2009; Mirme & Mirme, 2013; Manninen et al., 2016)
manufactured by Airel Ltd both in Hyytiälä and at CLOUD. The NAIS consists of two parallel differential mobility analyzers to measure the mobility distribution of positive and negative ions simultaneously. Ions are classified according to their electrical mobility and their concentration is recorded by a set of ring-shaped electrometers. The NAIS measures small ions and charged particles in the $0.0013$–$3.2$ $cm^2V^{-1}s^{-1}$ mobility range (ca. 0.8–40 nm in mobility diameter). The instrument alternates between three different measurement modes: ions, total aerosol and offset (zero measurements) mode. In Hyytiälä,
the NAIS was located in the same place as the DMPS, but it samples from ca. 3 m height above the ground.

### 2.3 Mass Spectrometer: CI-API-TOF

Sulfuric acid and HOM concentrations were measured with a chemical ionization atmospheric pressure interface time-of-flight mass spectrometer (CI-API-TOF, Jokinen et al. (2012)), which was located on top of a 35 m high tower just above the container area where the DMA-train was located. Vertical differences for HOMs are typically minor at that measurement
site, such that the above canopy measurements can be regarded as representative enough for our near-ground growth estimates (Zha et al., 2018). However, a comparison to few available days of ground-based CI-API-TOF measurements during the campaign revealed lower $[HOM_{dim}]$ but similar $[HOM_{tot}]$ pointing towards a significantly reduced transmission at large masses. The CI-API-TOF was equipped with a chemical ionization inlet with x-ray ionizer. Nitrate ion chemical ionization is a very selective method to detect strong acids, such as sulfuric acid and highly oxygenated organic compounds.
Calibration was performed using sulfuric acid (Kürten et al., 2012) and the obtained calibration coefficient ($2.6 \cdot 10^9$ $cm^{-3}$) was also used for the concentration measurement of HOM compounds. The total HOM concentration $[HOM_{tot}]$ was calculated as a sum of masses 260-622 Th. The HOM dimer concentrations $[HOM_{dim}]$ was retrieved from high-resolution peak fitting identifying the potential HOM dimers in the mass range above 400 Th, For CLOUD, a similar instrument was used and total HOM concentrations were estimated using a similar mass range and included a mass-dependent transmission
correction, which was not applied to Hyytiälä data due to missing calibrations. For CLOUD $[HOM_{dim}]$ includes onlynon-nitrate dimer peaks which could be identified (Lehtipalo et al., 2018).

### 2.4 Theoretical approaches for growth rate calculation

The neutral and ion growth rates have been calculated from atmospheric particle and ion size distribution with two different methods: maximum concentration and appearance time method. These two methods used in this study determine the time
when the growing mode reaches different diameters according to different criteria (Dada et al., 2020). The first method estimates the maximum concentration for the different mobility diameters and estimates the growth rate from a linear fit of these maximum concentration times versus diameter in the corresponding size range (Hirsikko et al., 2005). The second method, the appearance time method (Lehtipalo et al., 2014), estimates the 50% appearance time of the different mobility

diameters during a NPF event. It is important to note that both approaches estimate an apparent growth rate from the evolution of the particle size-distribution, which cannot necessarily be translated into a pure condensational growth rate of a single aerosol particle within that population. Population dynamic effects such as self-coagulation, extra-modal coagulation, cluster-contribution and changing vapor concentrations can all significantly influence the results of such apparent growth rate methods (Stolzenburg et al., 2005; Kontkanen et al., 2016; Li & McMurry, 2018; Olenius et al., 2014). However, the effect of coagulation is normally relatively small in Hyytiälä due to the moderate formation and sink rates (Kulmala et al., 2013). Therefore, methods which aim to disentangle these effects (e.g. Pichelstorfer et al., 2018) do not need to be applied and more importantly they could not be used with ion size distributions, due to the additional interactions between the ion and neutral particles (Leppä et al., 2011).

## 2.5 Aerosol and ion dynamics simulation with ion-UHMA

We use the University of Helsinki Multicomponent Aerosol model for neutral and charged particles (ion-UHMA) to simulate the basic dynamical processes (i.e. condensation, coagulation and deposition) as well as ion dynamics, i.e. ion-aerosol interaction and ion-ion recombination, during NPF. The ion-UHMA is a sectional model composed of 60 sections from 1.8 to 1000 nm, which include a neutral, positively charged and negatively charged population and their interactions. Sub-1.8 nm charged clusters are treated dynamically in the model, with an ion-pair production rate of 3 $cm^{-3}$ $s^{-1}$. The nucleation rates (neutral $J_n$, positive $J_+$ and negative $J_-$) are treated as input and are not determined by the model. Particle growth due to vapor condensation is calculated from the kinetically-limited condensation of sulfuric acid and a nano-Köhler type activation of the clusters by organics (Kulmala et al., 2004b). The collision efficiencies also consider charge and dipole effects (Nadykto and Yu, 2003; Stolzenburg et al. 2020). More details can be found in Leppä et al. (2009).

We performed three different simulations illustrating the importance of ion-processes in new particle growth. For two simulations we choose a setting representative for Hyytiälä with a diurnal pattern for condensable vapors, where in the first simulation (Hyytiälä-diurnal) we apply a diurnal nucleation rate following a sinusoidal profile as given in Eq. (5), which is the same approach as used by Leppä et al. (2009):

$$J_{1.7}[cm^{-3}\ s^{-1}] = J_{max} \cdot \sin\left(\frac{\pi}{2}\left(\frac{t-t_{start}}{t_{max}-t_{start}}\right)\right) \text{ for } t > t_{start} \text{ and } t < 2t_{max} - t_{start} \text{ else } 0 \qquad \text{Eq. (5)}$$

For the ion-induced nucleation $J_{1.7}^{diurnal}(\text{ion})$ we use $J_{max} = 1.0\ cm^{-3}\ s^{-1}$ and $t_{start} = 7\ h$, $t_{max} = 13\ h$, while for the neutral nucleation rate we use $J_{1.7}^{diurnal}(\text{neutral})$ we use $J_{max} = 1.0\ cm^{-3}\ s^{-1}$ and $t_{start} = 8\ h$, $t_{max} = 13\ h$. In the second simulation (Hyytiälä-parametrization) we follow the parametrization by Lehtipalo et al. (2018) (Eq. (2) and Eq. (4)) for the nucleation rate, but use an analytical idealization of the diurnal nucleation rate pattern as input for ion-UHMA. For the third setting (CLOUD-parametrization), we simulated the conditions in the CLOUD experiment, i.e. no background aerosol but wall losses and a different temporal behavior of the condensing vapors, but again following the parametrization by Lehtipalo et al. (2018) for the input nucleation rate within an analytical approximation for its temporal behaviour as input for ion-UHMA. The main parameters of the three model setups are also summarized in Table 1.

## 3. Results

### 3.1 Comparison between different approaches for growth rate calculation

Figure 1 compares the apparent growth rates using either total or ion size-distribution for growth rate analysis for two size-ranges (1.8-3.2 nm, Fig 1a; 3.2-8 nm Fig 1b) for our dataset from Hyytiälä. The applied analysis method does not result in significant systematic differences between the obtained growth rates as shown in Fig. S1, as the large majority of the measured GR are included in the [1:2; 2:1] range and the methods correlate rather well with an $R^2$ of 0.64 (1.8-3.2nm) and 0.47 (3.2-8nm). This corresponds well with earlier analysis of the differences between GR analysis methods (Yli-Juuti et al. 2011). In contrast, when we compare the results obtained by the same method, but using the total and charged particle size distributions, we see a significant offset towards lower ion GR values independent of the chosen method for our smaller size-interval (1.8-3.2 nm, Fig.1a), but not for the larger size range (3.2-8 nm, Fig. 1b). The same observation is also obtained when using the same instrument (i.e. NAIS) for the total and ion growth rate calculation (Supporting Information, Fig. S2), where however the total growth rate has generally higher uncertainties due to lower signal when compared to the DMA-train (see Kangasluoma et al., 2020) used for Fig.1. While the observed scatter in Fig. 1 is in the same range as obtained for the method comparisons (mainly within the 2:1/1:2 range, see Fig. S1), the ion GRs have a factor of 2 lower values on average than the total GRs. Altogether, these results demonstrate that the apparent (both maximum concentration and appearance time derived) ion and total particle growth cannot be viewed interchangeably below 3 nm.

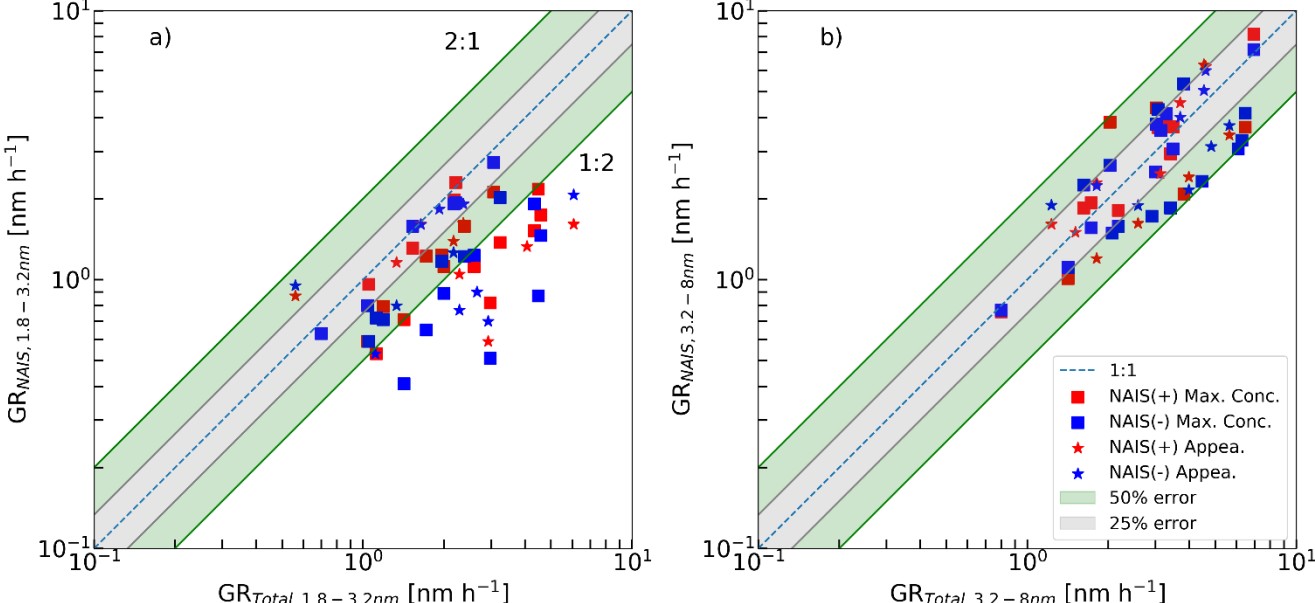

**Figure 1: Comparison of the apparent (maximum concentration, square symbols, and appearance time method, star symbols) ion GRs (GR_NAIS) and the total GRs (neutral plus charged, GR_Total measured with a DMA-train) for the dataset recorded between March-September 2020 in Hyytiälä. The blue dashed line shows the 1:1 ratio and the colored areas the 25% (grey) and 50% (green) deviation regions. Panel a compares the sub-3nm size-range (1.8-3.2 nm) and panel b the 3.2-8 nm size-range. Red symbols**

correspond to a measurement of the positive ion growth rate and blue symbols correspond to the measurement of the negative ion growth rates.

### 3.2 Growth rate comparison between total particles and ions observed in Hyytiälä and the CLOUD chamber

To explore further the discrepancy between apparent ion and total particle growth rates, we compare our results to measurements at the CERN CLOUD experiment. Earlier, Stolzenburg et al. (2020) and Lehtipalo et al. (2016) showed that the initial ion growth (below 3 nm) proceeds faster than total growth in the chamber when sulfuric acid and ammonia or sulfuric acid and amines are the condensable vapors. This is in line with the theoretical expectation that the polar sulfuric acid molecules exhibit an increased collision cross-section with charged particles due to dipole-charge interactions (e.g.

Nadykto & Yu, 2003). These results are contradictory to our ambient observations, but neither of the systems represent the conditions typical for the boreal forest. Therefore, we compare our results with the experiments with a mixture of $SO_2$, alpha-pinene, delta-3-carene, $O_3$, $NO_x$ and $NH_3$, which are more representative for Hyytiälä (Lehtipalo et al., 2018). Figure 2 shows a comparison between our ambient results from Hyytiälä and the CLOUD experiments. When looking at the ratio between ion and total population growth rates, we observe a clear difference between Hyytiälä and CLOUD. The sub-3 nm

ambient ion growth rates are on average clearly lower than the total growth rates (Fig. 2a), which is not reproduced in CLOUD (Fig 2c). However, at larger sizes (3-8 nm), both ambient measurements show no significant differences between the apparent ion and total growth rates (Fig. 2b), and the laboratory results even show slightly higher ion than total GRs (Fig. 2d), however the ion medians are not higher than the 75-quantile of the measured total GRs and therefore this effect could be well within potential statistical fluctuations. Fig. 2e confirms that the slower ion growth in the ambient measurement is

independent of the condensable vapor concentration as we plot the measured growth rates versus the modelled ones, calculated using the parametrization of Eq. (3) based on measured vapor concentrations. For Hyytiälä, we used the measured condensable vapor concentrations during the growth period ($NH_3$ was not measured but approximated by 150 pptv, see e.g. Makkonen et al., 2014). The CLOUD results are on the 1:1 line, as they are the basis for the parametrization and show no significant difference between the ion and total particle growth rates. The ambient growth rates are slightly higher than

predicted by the model, but still show a reasonable correlation with the modelled GRs. The higher measured GRs can be explained by uncertainties in the vapor concentration measurements and by the fact, that the parametrization does not

consider other organic precursors than HOM dimers, which probably leads to an underestimation.

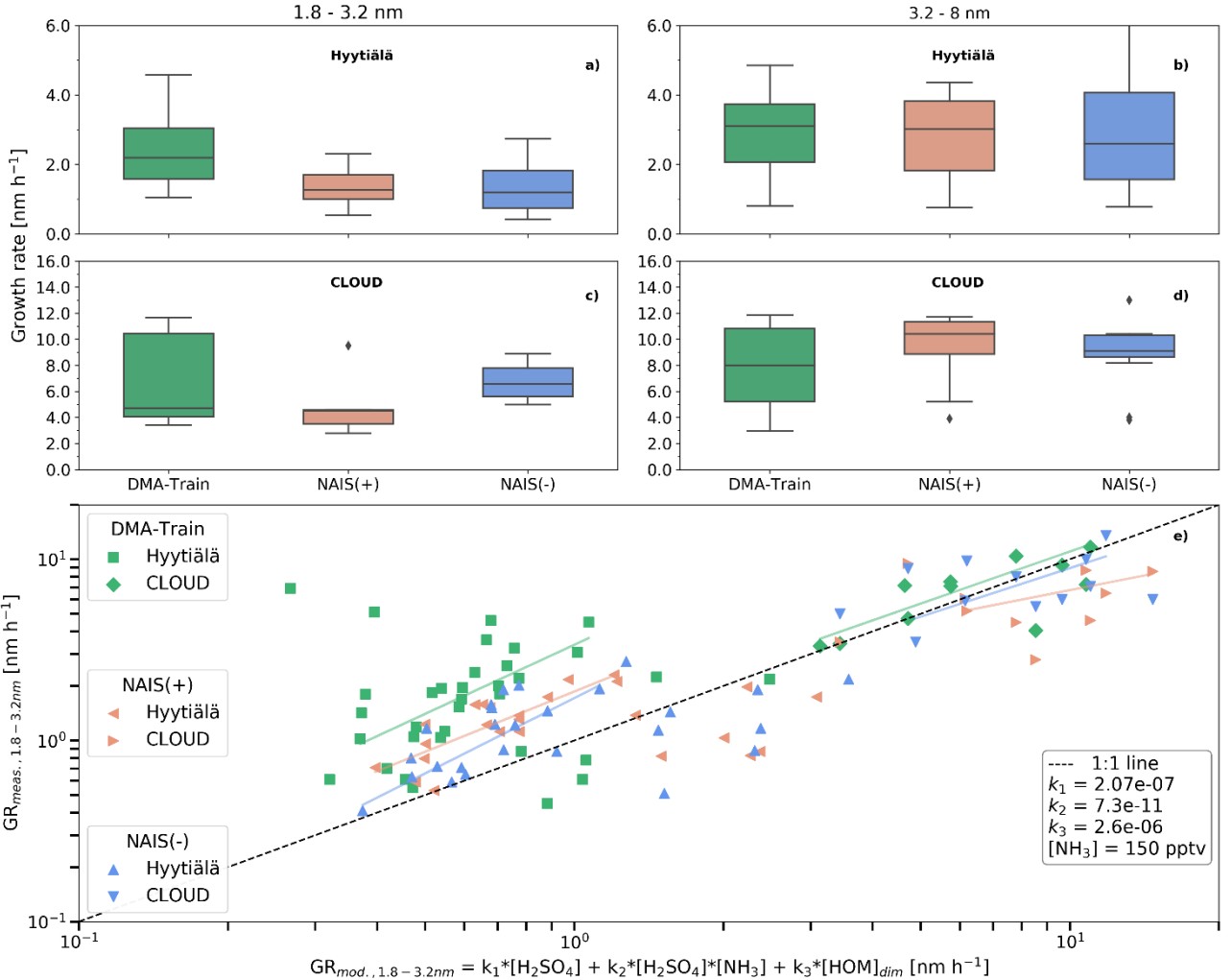

**Figure 2:** Comparison between GRs from CLOUD (Lehtipalo et al., 2018) and Hyytiälä. Due to higher vapor concentrations at CLOUD, the overall GRs are somewhat higher in CLOUD than in Hyytiälä. We represent the same instruments data with the same color code (i.e. green for DMA-Train, salmon for NAIS positive mode and blue for NAIS negative mode). Upper panels (a to d) are box plots of the growth rate distribution during Hyytiälä ("strong" NPF events only) and CLOUD campaign, showing the median growth rate and 25-75 percentiles of the data as horizontal black line and boxes and the 5-95 percentiles as errorbars and strong outliers as diamonds. Lower panel (e) shows all available growth rate measurements during both campaigns in relation to the total condensing vapour and their corresponding growth rate according to Eq. (3), with the green squares as the DMA-train data from Hyytiälä and green diamonds the DMA-train data from CLOUD NAIS growth rate data are displayed as triangles, with the orientation indicating if Hyytiälä or CLOUD data are used.

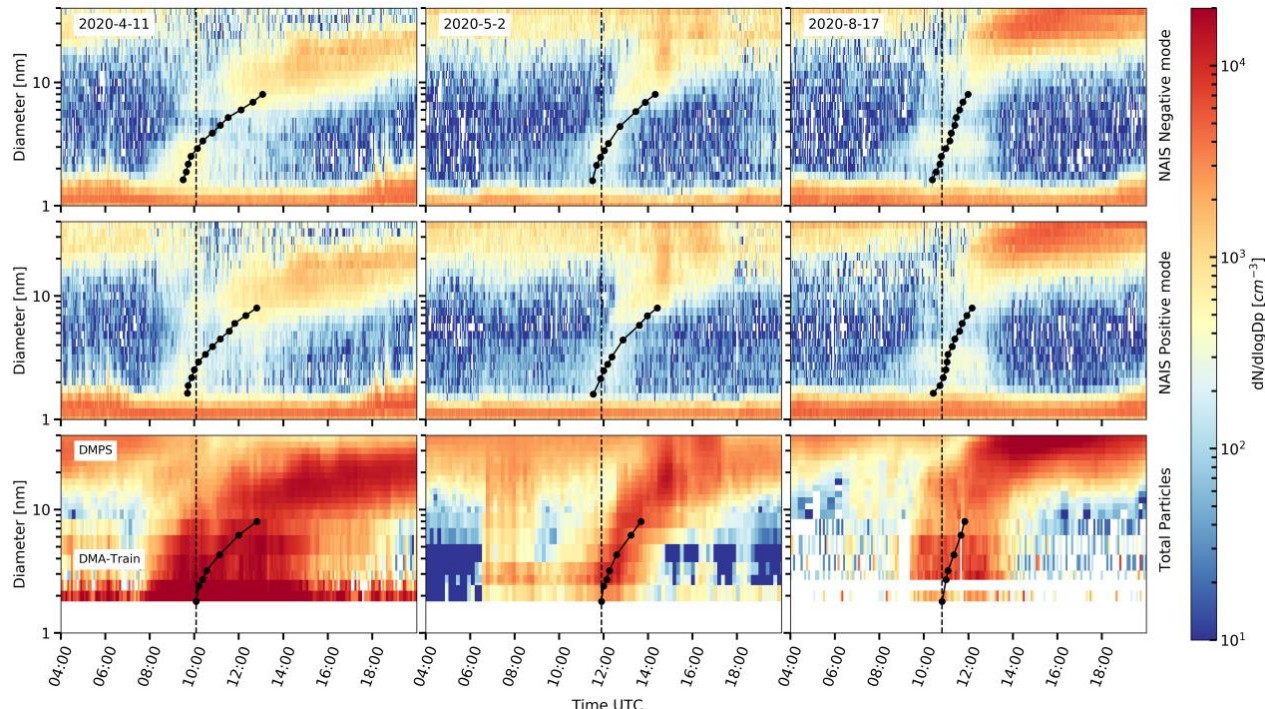

**Figure 3:** Evolution of the ion size-distribution (1st and 2nd row) measured with the NAIS and the combined total (neutral plus charged) size distribution measured with the DMA-train and the twin-DMPS during three characteristic events. The maximum diameter detected by the DMPS is cut off at 40 nm to have an easier comparison with the ion size distribution range of the NAIS. The black points represent the maximum concentration times for the different mobility diameters used to calculate the growth rate in the sub-10nm size ranges (see Supporting Information Fig. S3 for example fits of the maximum concentration time at 1.8 nm).

We investigate the dynamic behavior of the growth process in Fig. 3, where we present the total particle- and ion-size distribution during three characteristic NPF event days observed in Hyytiälä between March and September 2020. During the entire measurement period (spring to summer) the times of maximum concentration of the smallest ions (1.8-3 nm) during NPF events occur earlier (roughly 30-60 min) than the times of maximum concentration of the total particles of same size (see also Supporting Information Fig. S3 for the same observation with the appearance time method). As the concentration of ions is typically more than a factor of 10 lower compared to the total particle concentration, the earlier appearance of the ions during NPF has no significant effect on the appearance of the total growing mode (see also Stolzenburg et al., 2020). Therefore, this earlier appearance of the small ions results in a slower apparent ion growth rate compared to the total growth rate as the maximum concentration times of larger particles and ions agree better. This observation is in line with the results from Gonser et al. (2014) for a measurement site in Bavaria, Germany. Gonser et al. (2014) proposed a conceptual model to explain why we could observe faster total particle growth compared to ion growth in ambient measurements. If ion-induced nucleation starts earlier during daytime due to an increased cluster stability compared to the neutral pathway, the ion population will appear first. However, during the growth process, the growing ions are constantly neutralized by ion-ion recombination and the ion population is more influenced by charging of particles, which

are born neutral, with the latter becoming more and more significant when the neutral nucleation pathway becomes dominant. That way, the neutral and ion populations become indistinguishable at a later stage (also explaining the same ion and total particle growth rate for particles larger than 3 nm in both chamber and ambient experiments). The earlier appearance of the ion-population is therefore a possible reason for the reduced apparent ion growth rate, which is inferred by methods investigating the appearance of the population at a certain diameter. However, in the dataset of Gonser et al. (2014) the total size-distribution was limited to 2-2.5 nm and neither a quantitative understanding nor a supporting model for this effect were presented.

### 3.3 Ion-UHMA simulations of the particle-ion interaction

We tested the Gonser et al. (2014) conceptual model with aerosol dynamics simulations including ion processes (ion-UHMA, see Methods). In a first approach (Hyytiälä-diurnal simulations), we followed the basic arguments of Gonser et al. (2014) and used a simple sinusoidal diurnal pattern to describe the nucleation rate identical to the results of Leppä et al. (2009). In line with the conceptual model from Gonser et al. (2014), we implement that diurnal pattern for the nucleation rate with the 50% ion-induced fraction starting 1 hour earlier than the neutral nucleation. For the condensing vapors contributing to GR, we assumed that the sulfuric acid concentration has a diurnal sinusoidal variation between $1 \cdot 10^5$ and $4 \cdot 10^6$ cm$^{-3}$ (peak value close to the mean $H_2SO_4$ value of "strong" NPF events) and the nano-Köhler organics are constant at a concentration of $2 \cdot 10^7$ cm$^{-3}$ (same value as used in Leppä et al. (2009), close to the mean [HOM$_{tot}$] value of the "strong" NPF event). In the beginning of the simulation, two lognormal background particle modes are present, which are diluted following the diurnal pattern of an increasing boundary layer height after sunrise. The results are presented in Figure 4, where we can clearly observe a faster apparent particle growth rate for the total population compared to the ions (positive and negative) for the smallest size-interval, while there is no difference above 3 nm. However, it remains to be clarified, if we can justify the assumption of $J_{ion}$ starting earlier than $J_{tot}$ and if we can explain the absence of the slower ion growth rate in the chamber experiments.

As a second approach, we used the parametrization of the nucleation rates presented in Eq. (2) and Eq (4) as the basis for the nucleation rate input (Hyytiälä-parametrization). Figure **Error! Reference source not found.**5 shows the retrieval of the neutral and ion-induced J rates and the ion-induced fraction using that parametrization and the resulting diurnal variation in Hyytiälä based on the measured concentrations of sulfuric acid, ammonia and dimers of highly oxygenated molecules (HOM). The CLOUD results presented in Fig. 5a (Lehtipalo et al., 2018) show that both nucleation pathways (neutral and ion-induced) produce particles across all vapor concentrations, with the neutral nucleation rate scaling with increasing total nucleating vapor and the ion-induced nucleation being limited around the ion-pair production rate as already shown in Eq. (2) and Eq. (4) (Fig. 5a). Therefore, the fraction of ion-induced to total nucleation rate varies strongly with the available nucleating vapor concentrations, from almost 1 (below $10^{22}$ cm$^{-6}$ pptv) to almost 0 (above $10^{23}$ cm$^{-6}$ pptv) as can be seen in Fig. 5b. If the diurnal evolution of the total nucleating vapor concentrations crosses this vapor concentration range, we would obtain a situation where first the ion-induced nucleation and later the neutral pathway dominates the total nucleation rate.

Figure 5c illustrates that behavior for the measured data. However, the measured data needed to be scaled by a factor of 500 in order to be in the vapor concentration range where the transition between ion-induced and neutral dominated nucleation occurs in the related CLOUD experiments. This is in line with the higher observed than predicted GRs in Fig. 2e.

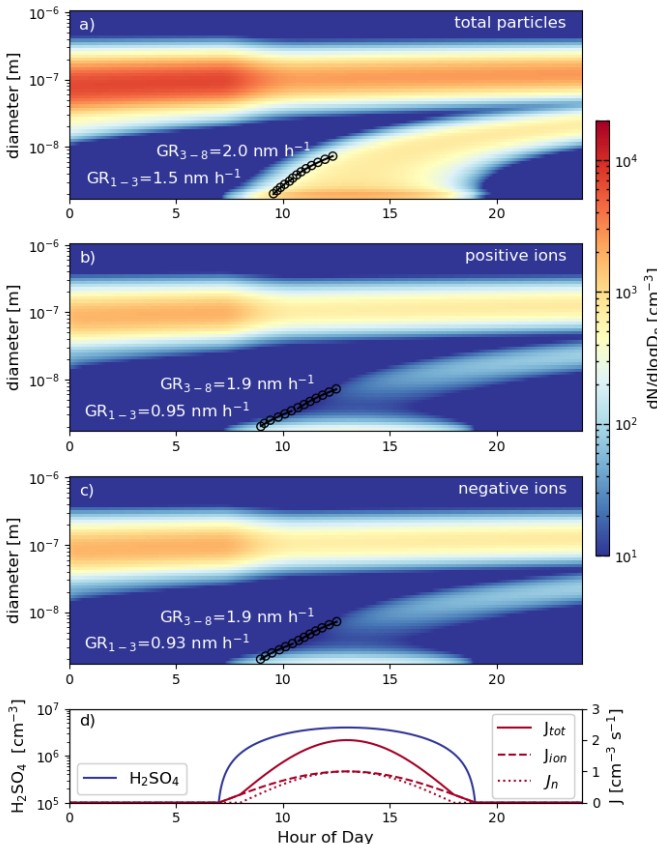

**Figure 4:** Model results based on Leppä et al. (2009) with the ion-induced nucleation starting 1 hour earlier than the neutral pathway as suggested by the conceptual approach of Gonser et al. (2014). The first panel corresponds to the total particle size distribution, the second and third panel are the ion (positive and negative) size distributions. The calculated apparent growth rate with the ion-UHMA model (black scatters) has been also plotted. The last panel correspond to the evolution of the nucleation rate of the total particle ($J_{tot}$) the ions ($J_{ion}$) and the neutral particles ($J_n$).

Apart from measurement uncertainties (around a factor of 2 for each vapor measurement, resulting in a potential offset of up to a factor $2^4=16$), there could be several other reasons why the critical range for the shift between ion-induced and total nucleation rate might be at lower concentration in Hyytiälä than CLOUD: 1) The parametrization from Lehtipalo et al. (2018) is based on CLOUD experiments, where higher cluster ion-concentrations (see also Table 1, 2000 versus 500) lead to more significant ion-induced nucleation (Wagner et al., 2017). Assuming a linear relationship between $N_{i,p}$ and $J$ (Dunne et al., 2016) would result in up to factor of 4 difference for the vapor concentration where the transition between ion-dominated and neutral-dominated occurs. 2) cluster stability is mostly controlled by ammonia, and therefore the importance of ion-induced nucleation might be strongly affected by ammonia availability and the importance of ammonia with respect to the

ion-induced fraction might be underestimated in that parametrization as ammonia concentrations were likely much lower in Hyytiälä than in most CLOUD runs (see Table 1). 3) Other factors than the nucleating vapor concentrations might also

crucially affect cluster stability and hence the fraction of ion-induced nucleation and the vapor range where the transition between ion dominated and neutral dominated nucleation occurs. Temperature and relative humidity could be crucial (Gagné et al., 2010) with especially the latter varying strongly between and during the different NPF event days (see Table 1), but were kept fixed in Lehtipalo et al. (2018). 4) We observe a significantly different [HOM$_{dim}$]/[HOM$_{tot}$] ratio for Hyytiälä and CLOUD (mean values "strong" NPF days 0.005 and 0.05, respectively), which could be caused by a reduced transmission of

the CI-API-TOF deployed in Hyytiälä (see Section 2.3). 5) It is not known which subset of oxidized organics are actually participating in the nucleation and growth processes in Hyytiälä. HOM dimer concentrations, chemical composition and volatility are very sensitive to the actual involved organic oxidation chemistry and temperature (Bianchi et al., 2019, Stolzenburg et al., 2018) and therefore significant differences between the chamber and ambient atmosphere are expected. Altogether, this leads to the conclusion that both the parametrization from Lehtipalo et al. (2018) is not perfectly transferable

to Hyytiälä conditions with respect to the importance of ion-induced nucleation and also our measured total condensable vapors especially [NH$_3$] and [HOM$_{dim}$] might also be underestimated.

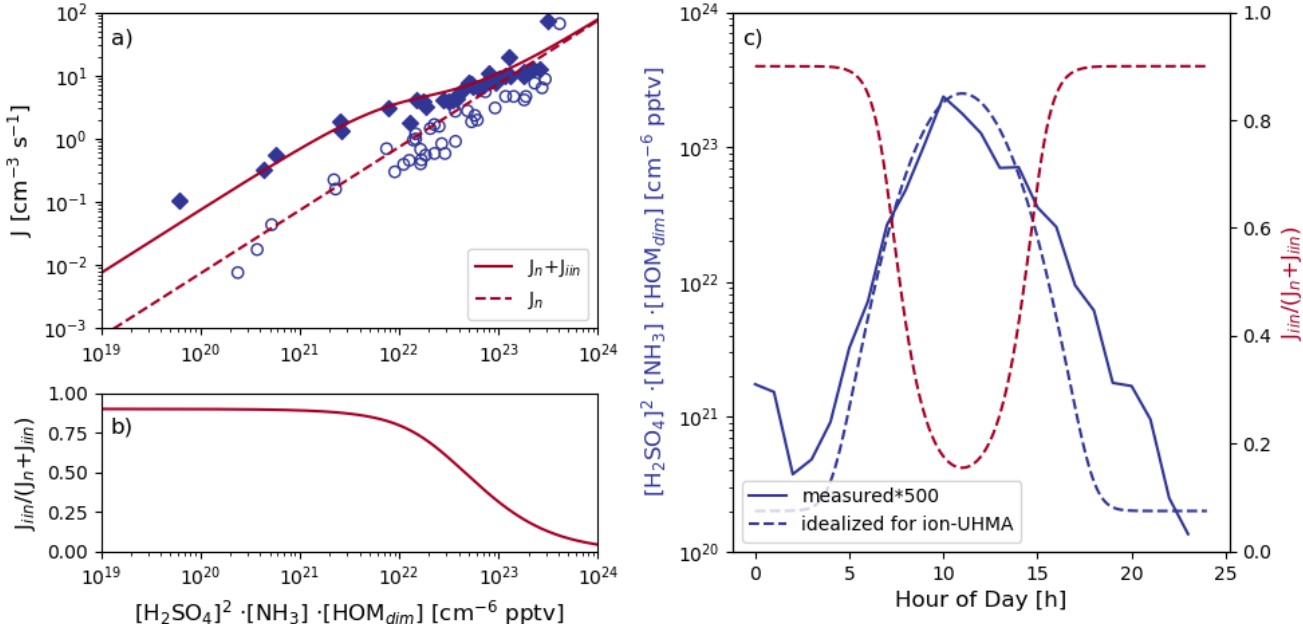

**Figure 5:** The importance of ion-induced nucleation in the boreal forest. a) shows the measured nucleation rates (diamonds under galactic cosmic ray conditions and circles under neutral conditions) and the parametrizations (Eq. (2) and Eq. (4)). b) shows the fraction of ion-
induced nucleation from total nucleation rate against the parametrizations from Eq, (2) and Eq. (4). c) shows the diurnal pattern of the calculated nucleation rates and ion-induced fraction in Hyytiälä.

Altogether, it is plausible that the transition from a high to a low ion-induced nucleation fraction in Hyytiälä happens at lower concentration than predicted by the parametrization. We therefore idealized the scaled nucleating vapor curve

(Fig. 5c) by an analytical Gaussian expression for computational simplicity and used it as input for the nucleation rate calculation with the parametrizations given in Eq. (2) and Eq. (4). We kept the vapors concentrations for the vapors contributing to growth ($H_2SO_4$ and nano-Köhler organics similar to $HOM_{tot}$) similar as for the Hyytiälä-diurnal simulation (with $H_2SO_4$ also changed to a Gaussian profile better resembling the actual diurnal pattern). The results are presented in Figure 6a-d and show a $GR_{ion} < GR_{tot}$ below 3 nm, but similar values above 3 nm identical to our ambient observations. In addition, the quantitative results are closer to the ambient measurements, confirming that it is indeed that slow transition from ion-induced to neutral dominated nucleation rate, which is responsible for the decreased apparent ion growth rate. In that sense, ion-induced nucleation is strictly speaking not starting one hour earlier as in the simple case but the increase of the vapor concentrations through the critical concentration range from almost unity to almost zero ion-induced fraction occurs within 3-5 hours in Hyytiälä (Fig. 5c). Moreover, the second simulation set also explains the absence of the effect at the CLOUD experiment: Here this transition occurs within ~10 minutes (sulfuric acid lifetime) in the CLOUD experiments, where all other vapor concentrations were typically kept constant and the nucleation burst was induced by switching on the UV lights inside the chamber and the subsequent formation of sulfuric acid. We show the results from simulations using such a vapor concentration profile together with adjusted boundary conditions (no background aerosol, but wall losses included) in Fig. 6e-h (CLOUD-parametrization). No significant difference is observed between the apparent total and ion growth rates for such simulation case. Note, that in simulations where sulfuric acid would be the major growth contributor, the dipole-charge interactions would even lead to a significantly enhanced ion growth rate compared to the total growth rate. However, in the experiments by Lehtipalo et al. (2018), the organics are dominating the growth and no dipole-charge interactions are considered for the collisions of organics with growing ions in the simulations.

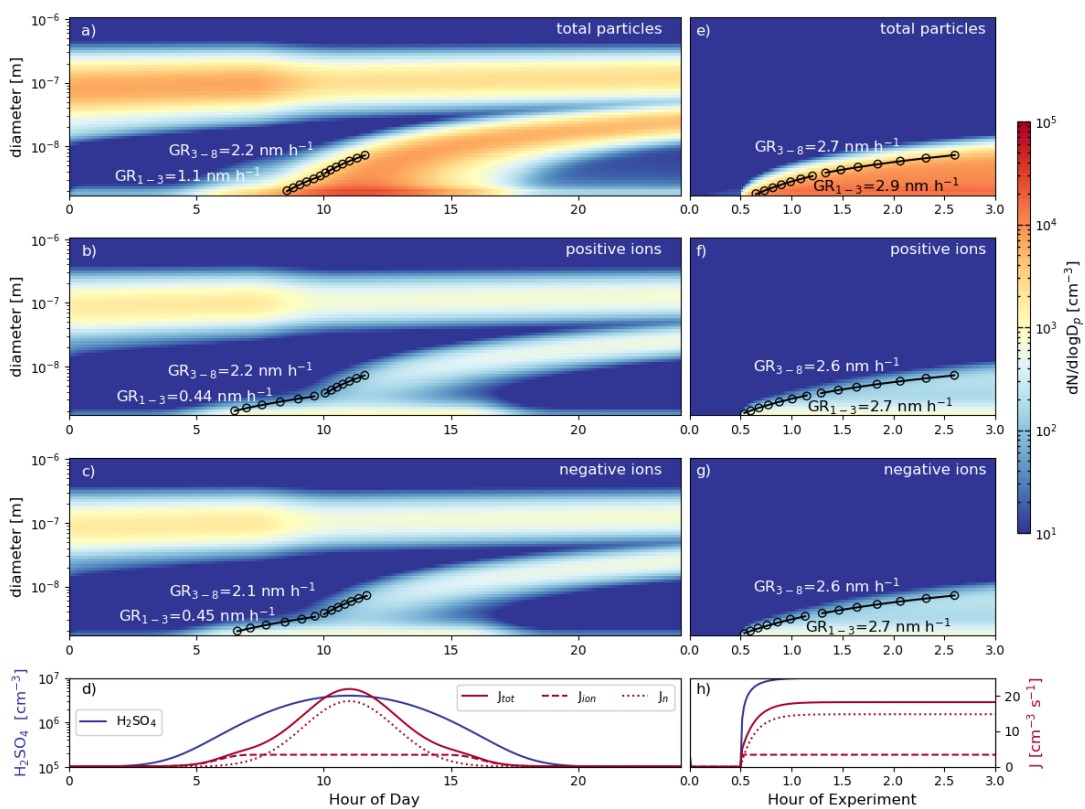

**Figure 6:** Ion-UHMA model results with the J rates parametrized according to Lehtipalo et al. (2018) for both Hyytiälä (Panels a-d) and CLOUD (Panels e-h) conditions. The upper panels show the total particle size-distribution, the second and third row panels the positive and negative ion size distribution, respectively. The last row panels show the sulfuric acid concentration for vapor growth (blue) and the parametrized nucleation rate according to Lehtipalo et al. (2018) assuming typical profiles of $[H_2SO_4]^2$ $[NH_3]$ $[HOM_{dim}]$ for Hyytiälä and CLOUD.

## 4. Conclusions

The role of ions in atmospheric new particle formation and initial growth is still not fully clarified, although the ion populations are often used to infer nanoparticle growth rates. We have shown that apparent particle growth rates in the sub-3 nm range can be underestimated if ion size-distributions are used instead of total size distributions. We observed, during the entire period of measurement in the boreal forest (Spring to Summer 2020), an earlier formation of ions than total particles in the sub-3nm range. As typical ion concentrations are a factor 10 less compared to total particle concentration, the earlier ion appearance did not affect the appearance of the total growing mode, but resulted in slower apparent ion growth rates. Previous work suggests that in the case of condensing polar molecules such as sulfuric acid, the growth of sub-3 nm charged particles should be enhanced compared to the neutral particle growth (Stolzenburg et al., 2020). However, the mix of condensable vapors in Hyytiälä is more complex and therefore we compared the observation in Hyytiälä with results from CLOUD chamber experiments under similar conditions (Lehtipalo et al., 2018). While the parametrization from the chamber

experiments can reasonably predict the observed order of magnitude of the ambient growth rates, we observe no difference between total and ion growth rate in contrast to our ambient observations.

Gonser et al. (2014) proposed a conceptual model to explain the observation of slower ion growth where ion-induced nucleation start earlier during day-time but during growth process ions are constantly neutralized by ion-ion recombination. That way, neutral and ion population become very difficult to distinguish at larger sizes, but this results in slower apparent ion growth rates for sub-3 nm sizes. Here, we confirmed the conceptual model with aerosol dynamics simulations based on the ion-UHMA model, which includes neutral, positively and negatively charged populations and their interactions. We modelled the nucleation rate according to Lehtipalo et al. (2018) and showed quantitatively how the transition from an ion-induced dominated nucleation regime to a neutral dominated nucleation scheme leads to apparent sub-3 nm ion growth rates, which are roughly a factor of 2 lower than the total growth rate, in good agreement with our ambient observations. The simulations also provided the explanation of the absence of this effect during CLOUD measurements, where the nucleating vapor concentrations are typically changed within 10 minutes and hence the change from ion-induced nucleation into the neutral dominated nucleation occurs much faster than in Hyytiälä. Altogether, our results show that the apparent (i.e. maximum concentration or appearance time method based) ion GR do not correspond to the real condensational growth (also not the combined condensational and coagulation growth) of the particle population, but are heavily affected by the temporal behavior of ion-induced nucleation and aerosol-ion dynamics processes like ion-ion recombination and particle diffusion charging. Sub 3-nm apparent growth rates based on ion population measurements are therefore not necessarily suited to infer information on the abundance of condensable vapors or their seasonal variation and should always be interpreted cautiously. However, the effect of reduced ion GR compared to total particle GR depends on the actual nucleation mechanism, the abundance of ions and ultimately on the transition time between an ion-induced and neutral dominated nucleation regime and hence it is difficult to transfer our findings to other environments. In settings where ion-induced nucleation always dominates (e.g. at remote or high-altitude sites) or where the transition is very short due to a strong neutral nucleation pathway (e.g. polluted settings with strong $H_2SO_4$-amine clustering), we expect to see less differences between ion and total GR.

**Data availability**

The current version of the ion-UHMA simulations are available from the GitLab repository https://version.helsinki.fi/atm/ion-uhma/-/tree/hyde-ion-gr and the version upon publication of this manuscript is archived under https://doi.org/10.23729/328f4a5c-006f-4563-a0cc-2728bda6ef4c

## Conflict of Interest

The authors declare that they have no conflict of interest.

## Author contributions

D.S. designed the study, L.G.C., K.L., L.R.A., N.S., S.H., J.K., D.S. performed the measurements, D.S. performed the simulations, L.G.C., K.L., N.S., D.S. analyzed the data, L.G.C., K.L., J.K., M.K., P.M.W., D.S. were involved in the scientific discussion and interpretation of the results, L.G.C. and D.S. wrote the manuscript, all authors commented and edited the final manuscript.

## Acknowledgements

We acknowledge the following projects: ACCC Flagship funded by the Academy of Finland grant number 337549, Academy professorship funded by the Academy of Finland (grant no. 302958), Academy of Finland projects no. 1325656, 316114 and 325647; University of Helsinki 3-year grant (75284132), Russian Mega Grant project "Megapolis - heat and pollution island: interdisciplinary hydroclimatic, geochemical and ecological analysis (075-15-2021-574); "Quantifying carbon sink, CarbonSink+ and their interaction with air quality" INAR project funded by Jane and Aatos Erkko Foundation, European Research Council (ERC) project ATM-GTP Contract No. 742206, Samsung $PM_{2.5}$ SRP, the European Union's Horizon 2020 research and innovation programme Marie Sklodowska-Curie grant agreement no. 895875 ("NPF-PANDA") and the Marie Skłodowska Curie ITN "CLOUD-MOTION" (764991). Technical and scientific staff in Hyytiälä are acknowledged.

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
