# Peer review of "On the relation between apparent ion and total particle growth rates in the boreal forest and related chamber experiments"

_Atmospheric Chemistry and Physics, 2022_

## Referee Comment (RC2)

The study by Gonzalez Carracedo et al. reports observations from the boreal forest that show that below 3 nm, the particle growth rate derived from ion observations is lower than that derived from total particle concentration measurements. This work confirms observations from a previous study (Gonser et al., 2014) and provides an explanation, through the combined analysis of real-atmosphere and laboratory measurements and the use of a process model. Because of its indirect impact on climate, NPF is a key atmospheric process whose study is crucial; the question addressed in this study is of particular interest since the possibility of generalizing observations made from ion measurements (often easier and more direct than neutral particle measurements) to the total particle population is critical. This is an otherwise well written paper, which I therefore recommend for publication in ACP. I suggest however that the comments listed below be addressed before final publication, with, in particular, a clarification in the description of the simulations that are at the heart of this study (Sects. 2.5 and 3.3).

Comment 1: P1, L29 : I think that a complementary reference to the more recent reviews by Kerminen et al. (2018) or Lee et al. (2019) could shed light on observations made during the last ~20 years, after the review by Kulmala et al. (2004) was published.

Comment 2: P2, L41-42: I would suggest referring to papers that describe the most commonly used methods for GR calculation at this point in the text; in addition to the "maximum" (Hirsikko et al., 2005) and appearance time (Lehtipalo et al., 2014) approaches which are more fully described later in the paper, the work of Dal Maso et al. (2005) could for example be mentioned.

Comment 3: P2, L44: In addition to Manninen et al. (2009), I would suggest referring here to the studies by Mirme and Mirme (2013) and Manninen et al. (2016) which are more dedicated to the description of the instrument and its functioning, and which are mentioned elsewhere in the text. Manninen et al. (2016) could also be added at L109.

Comment 4: P2, Sect. 2.1: It would be interesting to indicate / recall (here or later in the results Sect.) how many NPF events were considered in the analysis, for both Hyytiälä and CLOUD datasets.

Comment 5: P3, Table 1:

- If I am not mistaken, the names given to the simulations (according to the formulation used for J) are only used in Table 1, whereas their use in the text could help when describing successively the different simulations. I think that it should also be made clear that "simple" refers to the prescription of a diurnal profile of J, whereas "lehtipalo" indicates the use of parameterizations;
- Why is it indicated "not used" for the HOM dimers concentration while, in line with the formulation of the parameterizations of J (Eqs. 2 and 4), this variable is a priori used in the simulations (e.g.: L251-252)?
- Why is there no value for the total HOM concentration in the CLOUD experiments?
- In the line indicating the product of the precursor concentrations used in Eqs. 2 and 4, [HOM] should be replaced by [$HOM_{dim}$], right?
- The meaning of $Q_{i.p.}$ and $N_{i.p.}$ should be explained.

I think that in general, in relation to Sect. 3.3, a clearer description of the simulations would be beneficial to the understanding of the results. However, Sect. 2.5 may come too early in the manuscript for such a detailed description, so one option would be to keep only the general description of the model in Sect. 2.5 and offer a more complete/detailed description of the simulations directly in Sect. 3.3 (and in this case, the last three columns of Table 1 could be removed and made into a table of their own in Sect. 3.3).

Comment 6: P3, L84-85: "we also find a parametrization for the ion-induced nucleation fraction based on the ion-pair production rate and the vapor concentrations at which ion-induced nucleation becomes less dominant": the approach used to derive the parameterization is not clear to me, can the authors try to specify / clarify?

Comment 7: P4, L86: The equation number should be corrected (4 instead of 3).

Comment 8: P4, L94-96: "a good sensitivity towards low particle concentrations in the sub-10 nm range", "the DMA-train can measure also sub-3 nm particle growth with an unprecedented sizing precision": Is there a study in which the performance of the DMA-train has been compared with that of instruments that measure over a comparable size range, including in particular NAIS to also support L169-170 ("where however the total growth rate has generally higher uncertainties due to lower signal when compared to the DMA-train")?

Comment 9: P5, Sect. 2.3:

- Since the mass spectrometry measurements were conducted at a different height than the other measurements (which may therefore raise questions about potentially different conditions with respect to forest canopy), I would suggest referring to Zha et al. (2018), who indicate that during daytime, i.e. the period of interest for NPF, HOM measurements (concentrations and composition distributions) are in fact similar below (i.e. near the ground) and above (i.e. at 35 m height) the canopy;
- I would suggest saying few words on the calibration factor used to convert CI-APi-TOF signals into concentrations in molecules / $cm^3$. At least, was the same coefficient applied to sulfuric acid and HOM signals?
- L125: "at similar instrument" → a similar instrument
- L126: "Lehtiplao" → Lehtipalo

Comment 10: P5, L136: "which cannot necessarily translated into a pure condensational growth rate": the wording should be checked.

Comment 11: P6, L166-167: I take the opportunity of this first illustration of a result that is at the heart of this study to make a more general comment. For a given instrument and calculation method, $GR_{Ions}$ appears to be indeed lower than $GR_{Total}$ below 3 nm. I think it would be interesting to try to quantify (at least roughly) these differences (there is a factor of 2 indicated in the conclusions L338; it would be interesting to mention and discuss this earlier in the manuscript), especially to quickly discuss their magnitude/importance in relation to the uncertainties on the GR estimate related to the different calculation methods and/or the use of different instruments (e.g. Yli-Juuti et al., 2011). In other words, is the estimation error made on $GR_{Total}$ considering $GR_{Ion}$ of the same order, lower, higher, than the differences in the evaluation of $GR_{Total}$ related to the use of different methods / instruments?

Comment 12: P6, L169: In order to avoid any confusion, I would clearly refer to NAIS: "when using the same instrument (i.e. NAIS)…".

Comment 13: P7, Fig. 1.b: Similar to Fig. 1.a, I would suggest adding the lines delimiting the [1:2, 2:1] range to help in the visual evaluation of the differences between $GR_{Ions}$ and $GR_{Total}$.

Comment 14: P7, L191: Since the ranges delimited by the boxes / error bars overlap in Fig. 2.a, I would suggest saying "The sub-3 nm ambient ion growth rates are on average clearly lower than the total growth rates…".

Comment 15: P7, L192-193: "However, at larger sizes (3-8 nm), both the laboratory and ambient measurements show no significant differences between the apparent ion and total growth rates (Fig. 2b and 2d)": I do not agree with this statement; the GRs obtained from ion measurements in CLOUD (Fig. 2.d) are on average higher than the total GR, with differences that appear to be of the same order as those found for the ambient measurements in the lower size range (Fig. 2.a).

Comment 16: P8, Fig. 2:

- Fig. 2.a-d: the meaning of the symbol should be defined explicitly (median, quartiles, range indicated by the error bars, signification of the diamonds);
- Fig. 2.e: In order to ease the reading of the figure, I would suggest keeping the same symbol for Hyytiälä and CLOUD for all instruments, and only use different colours to distinguish between the different instruments.

Comment 17: P10, L230-231: "with the latter becoming more and more significnat at a later stage when also the neutral nucleation pathway dominant": the wording should be checked.

Comment 18: P10, L245: "The results are presented in Figure we can clearly observe": the wording should be checked.

Comment 19: P10, Sect. 3.3: Related to Comment 5, I would suggest clarifying the description of the simulations, as well as of some results. Specific points that I would in particular suggest to address are listed below:

- Does the "organic concentration" mentioned in L241correspond to $HOM_{tot}$ in Table 1? If so, why are the values given in the text and in Table 1 different ($1\times10^7$ vs $2\times10^7$ $cm^{-3}$)? More broadly, as done for sulphuric acid in L241, the concentrations chosen for the other precursors should at some point be discussed and justified;
- It think it would help to clearly recall at L240-242 that the mentioned species are used for GR calculation in the model (to avoid any confusion with J);
- L249: Eq. (4) was used as well, wasn't it?

I also strongly suggest to clarify the description of Fig. 5 (expanding / clarifying in particular the sentence that is currently in lines 250-252):

- indicating clearly that the measurements reported on Fig. 5.a are those from the CLOUD experiments which allowed to derive the parametrizations of J used in the simulation;
- specifying how the diurnal profile of the product of the concentration of measured precursors is obtained (average over NPF event days at the station?);
- indicating perhaps also clearly that the profile of the fraction of ion-induced nucleation represented in Fig 5.c is obtained by considering the idealized profile of the product of the concentration of precursors shown in the same figure.

Comment 20: P12, L282: udnerestimated → underestimated

Comment 21: P13, Fig. 6: the diurnal profile of sulfuric acid concentration shown in Figure 6.d appears to be different from that shown in Figure 4.d, and unless I am mistaken it is not indicated in the text that the sulfuric acid profile is different in the two simulations. Can the authors provide an explanation?

Comment 22: P14, Conclusions : This study is dedicated to the boreal forest but do we have any idea / can we anticipate the observations that could be made in other environments and in particular at high altitude, where the role of ions in nucleation is a priori more marked than at low altitude (Manninen et al., 2010; Sellegri et al., 2019)? Can we assume that we have less error in $GR_{total}$ based on ion observations there, or is it difficult to anticipate without detailed knowledge of the precursors involved?

References :

Dal Maso, M., Kulmala M., Riipinen, I., Wagner, R., Hussein, T., Aalto, P. P., and Lehtinen, K. E. J.: Formation and growth of freshatmospheric aerosols: eight years of aerosol size distribution data from SMEAR II, Hyytial̈ a, Finland, Boreal Environ. Res., 10, 323–336, 2005.

Kerminen, V.-M., Chen, X., Vakkari, V., Petäjä, T., Kulmala, M., and Bianchi, F.: Atmospheric new particle formation and growth: review of field observations, Environ. Res. Lett., 13, 103003, https://doi.org/10.1088/1748-9326/aadf3c, 2018.

Lee, S.-H., Gordon, H., Yu, H., Lehtipalo, K., Haley, R., Li, Y., & Zhang, R. (2019). New particle formation in the atmosphere: From molecular clusters to global climate. Journal of Geophysical Research: Atmospheres, 124, 7098– 7146. https://doi.org/10.1029/2018JD029356.

Sellegri, K., Rose, C., Marinoni, A., Lupi, A., Wiedensohler, A., Andrade, M., Bonasoni, P. and Laj, P. : New particle formation : a review of ground-based observations at Mountain research stations, Atmosphere, 10(9), 493, https://doi.org/10.3390/atmos10090493, 2019.

Zha, Q., Yan, C., Junninen, H., Riva, M., Sarnela, N., Aalto, J., Quéléver, L., Schallhart, S., Dada, L., Heikkinen, L., Peräkylä, O., Zou, J., Rose, C., Wang, Y., Mammarella, I., Katul, G., Vesala, T., Worsnop, D. R., Kulmala, M., Petäjä, T., Bianchi, F., and Ehn, M.: Vertical characterization of highly oxygenated molecules (HOMs) below and above a boreal forest canopy, Atmos. Chem. Phys., 18, 17437–17450, https://doi.org/10.5194/acp-18-17437-2018, 2018.

---

## Author Comment (AC1)

**Reviewer #1:**

Carracedo and co-authors present field and laboratory measurements of apparent ion and total particle growth rates, and compare them with aerosol dynamics simulations using the ion-UHMA model. Specifically, they evaluate ion and particle growth rates in the size range from 1.8 to 3.2 nm and in the size range from 3.2 to 8 nm observed under ambient conditons at the Hyytiälä field site in 2020, and from CLOUD experiments simulating Hyyiälä conditions in a controlled chamber. Observed differences are convincingly explained by the ion-UHMA aerosol dynamics simulations when the ion-induced vs. neutral nucleation rates are parameterized as functions of the condensing vapor concentrations. This is also consistent with a previous conceptual model describing differences in ion and total growth rates in ambient measurements. In Hyytiälä, the transition from a high to a low relevance of ion-induced nucleation typically happens over several hours in the morning, while it happens within minutes in the CLOUD experiments. Overall, the manuscript is well-written, and I very much appreciate the diligent scientific discussion and interpretation of the presented measurements and simulations. The study is original, and I only have a few minor comments. I recommend publication in ACP after minor revisions.

We thank the reviewer for their insightful comments, and we have revised the manuscript accordingly. Please find our results in black and the changes in red.

Specific comments:

In section 2.5, the explanation of the three different simulations performed using ion-UHMA is difficult to understand. In my opinion, it would help if the J parameterizations used in the simulations were added in an additional line of Table 1. Also, the difference of the two simulations representative for Hyytiälä is difficult to understand in this section. Please extend the description of the simulations to clarify.

We agree with the reviewer and extend our description of the simulations in Section 2.5 and added additional lines to Table 1, which show the J values and the used equations for their parametrization. We also added a more detailed description how the input concentrations were chosen in Section 3.3 (see response to referee #2) and in line with the comments of referee #2, we agreed to change the names of the simulations and use them now consistently throughout the manuscript.

"We performed three different simulations illustrating the importance of ion-processes in new particle growth. For two simulations we choose a setting representative for Hyytiälä with a diurnal pattern for condensable vapors, where in the first simulation (Hyytiälä-diurnal) we apply a diurnal nucleation rate following a sinusoidal profile as given in Eq. (5), which is the same approach as used by Leppä et al. (2009):

$$J_{1.7}[cm^{-3} \, s^{-1}] = J_{max} \cdot \sin\left(\frac{\pi}{2}\left(\frac{t-t_{start}}{t_{max}-t_{start}}\right)\right) \text{ for } t > t_{start} \text{ and } t < 2t_{max} - t_{start} \text{ else } 0 \qquad (5)$$

For the ion-induced nucleation $J_{1.7}^{diurnal}$(ion) we use $J_{max} = 1.0 \, cm^{-3} \, s^{-1}$ and $t_{start} = 7 \, h$, $t_{max} = 13 \, h$, while for the neutral nucleation rate we use $J_{1.7}^{diurnal}$(neutral) we use $J_{max} = 1.0 \, cm^{-3} \, s^{-1}$ and $t_{start} = 8 \, h$, $t_{max} = 13 \, h$. In the second simulation (Hyytiälä-parametrization) we follow the parametrization by Lehtipalo et al. (2018) (Eq. (2) and Eq. (4)) for the nucleation rate, but use an analytical idealization of the diurnal nucleation rate pattern as input for ion-UHMA. For the third setting (CLOUD-parametrization), we simulated the conditions in the CLOUD experiment, i.e. no background aerosol but wall losses and a different temporal behavior of the condensing vapors, but again following the parametrization by Lehtipalo et al. (2018) for the input nucleation rate within an analytical approximation for its temporal behaviour as input for ion-UHMA. The main parameters of the three model setups are also summarized in Table 1."

In section 3.1, in order to show the difference of apparent ion and total particle growth rates below and above 3 nm, in my opinion it would be more instructive to present Fig. 1b and Fig. S1 as two panels of a

revised Figure 1 in the main manuscript. The comparison of the two growth rate analysis methods is interesting but I recommend moving Fig. 1a to the supplementary material.

We agree with the reviewer and changed Figure 1 as suggested. Please not the revised paragraph describing the Figures:

"Figure 1 compares the apparent growth rates using either total or ion size-distribution for growth rate analysis for two size-ranges (1.8-3.2 nm, Fig 1a; 3.2-8 nm Fig 1b) for our dataset from Hyytiälä. The applied analysis method does not result in significant systematic differences between the obtained growth rates as shown in Fig. S1, as the large majority of the measured GR are included in the [1:2; 2:1] range and the methods correlate rather well with an $R^2$ of 0.64 (1.8-3.2nm) and 0.47 (3.2-8nm). This corresponds well with earlier analysis of the differences between GR analysis methods (Yli-Juuti et al. 2011). In contrast, when we compare the results obtained by the same method, but using the total and charged particle size distributions, we see a significant offset towards lower ion GR values independent of the chosen method for our smaller size-interval (1.8-3.2 nm, Fig.1a), but not for the larger size range (3.2-8 nm, Fig. 1b). The same observation is also obtained when using the same instrument (i.e. NAIS) for the total and ion growth rate calculation (Supporting Information, Fig. S2), where however the total growth rate has generally higher uncertainties due to lower signal when compared to the DMA-train (see Kangasluoma et al., 2020) used for Fig.1. While the observed scatter in Fig. 1 is in the same range as obtained for the method comparisons (mainly within the 2:1/1:2 range, see Fig. S1), the ion GRs have a factor of 2 lower values on average than the total GRs. Altogether, these results demonstrate that the apparent (both maximum concentration and appearance time derived) ion and total particle growth cannot be viewed interchangeably below 3 nm."

In section 3.3, the measured field data are scaled by a factor of 100, and possible reasons why this may be justified are qualitatively discussed. Would it be possible to give some indication which of the four specific reasons might be most important for this difference, or how important measurement uncertainties could be relative to the specific reasons?

Our total vapor concentrations used for the input in the nucleation rate parametrization were so far based on an individual day, but following the comments of referee #2, we decided to base all values on campaign averages. Therefore, the observed difference in total vapor concentration is even higher (factor of 500). We understand the reviewer's request for a more quantitative statement here, and due to the adjusted numbers we even more see the reason to roughly quantify if this difference can be explained quantitatively. However we feel that it would be pure speculation on some of the mentioned potential factors and hence we only give quantitative estimates were we are convinced that it is reasonable.

We added the typical measurement uncertainties of condensable vapor measurements to the text to illustrate that they can contribute easily a factor of 16 ($2^4$) in that context. Moreover, we added the numbers found in Wagner et al. (2017) for the cluster ion concentrations to the text, which (assuming linear behavior between $N_{i,p}$ and $J$ as in Dunne et al. (2016)) could contribute a factor of 4. We also note that the HOM_dim/HOM_tot ratio in Hyytiälä is 0.005 while it is on average 0.05 at CLOUD, indicating that our $HOM_{dim}$ concentrations might be largely underestimated in Hyytiälä due to missing transmission corrections or different dimer formation chemistry. This remarkable difference could cause at least a factor of 10 difference and together with the other points above (16*4*10=640) this is already enough to explain the observed differences.

For the other possible reasons (cluster stability due to RH, T and specific HOM composition) we refrain from any quantification of the induced offset due to our limited understanding of these aspects. Hence we also cannot conclude definitely which might be the biggest contributor to the observed difference, but we are confident that the added numbers give the reader enough information to estimate the magnitudes of the different effects. We adjusted the text as follows:

"Apart from measurement uncertainties (around a factor of 2 for each vapor measurement, resulting in a potential offset of up to a factor $2^4=16$), there could be several other reasons why the critical range for the shift between ion-induced and total nucleation rate might be at lower concentration in Hyytiälä than CLOUD: 1) The parametrization from Lehtipalo et al. (2018) is based on CLOUD experiments, where higher cluster ion-concentrations (see also Table 1, 2000 versus 500) lead to more significant ion-induced nucleation (Wagner et al., 2017). Assuming a linear relationship between $N_{(i,p)}$ and J (Dunne et al., 2016) would result in up to factor of 4 difference for the vapor concentration where the transition between ion-dominated and neutral-dominated occurs. 2) cluster stability is mostly controlled by ammonia, and therefore the importance of ion-induced nucleation might be strongly affected by ammonia availability and the importance of ammonia with respect to the ion-induced fraction might be underestimated in that parametrization as ammonia concentrations were likely much lower in Hyytiälä than in most CLOUD runs (see Table 1). 3) Other factors than the nucleating vapor concentrations might also crucially affect cluster stability and hence the fraction of ion-induced nucleation and the vapor range where the transition between ion dominated and neutral dominated nucleation occurs. Temperature and relative humidity could be crucial (Gagné et al., 2010) with especially the latter varying strongly between and during the different NPF event days (see Table 1), but were kept fixed in Lehtipalo et al. (2018). 4) We observe a significantly different $[HOM_{dim}]/[HOM_{tot}]$ ratio for Hyytiälä and CLOUD (mean values "strong" NPF days 0.005 and 0.05, respectively), which could be caused by a reduced transmission of the CI-API-TOF deployed in Hyytiälä (see Section 2.3).  5) It is not known which subset of oxidized organics are actually participating in the nucleation and growth processes in Hyytiälä. HOM dimer concentrations, chemical composition and volatility are very sensitive to the actual involved organic oxidation chemistry and temperature (Bianchi et al., 2019, Stolzenburg et al., 2018) and therefore significant differences between the chamber and ambient atmosphere are expected. Altogether, this leads to the conclusion that both the parametrization from Lehtipalo et al. (2018) is not perfectly transferable to Hyytiälä conditions with respect to the importance of ion-induced nucleation and also our measured total condensable vapors especially $[NH_3]$ and $[HOM_{dim}]$ might also be underestimated."

Technical comments:

We thank the reviewer for finding all those small typos in the text. We corrected all of them accordingly.

l.49: remove "J." in citation "Leppä et al."

adjusted.

l.72: In Table 1, please explain parameters Q_i,p and N_i,p. In the ion-UHMA columns, the labels "J_simple_ambient", "J_lehtipalo_ambient" and "J_lehtipalo_chamber" are not clear to me. Maybe simply label the columns "simulation 1, 2, 3" and add additional lines explaining the parameterization of J. To be consistent, add parentheses around asterisk after NH3 mixing ratios in Hyytiälä, 50-150(*).

We decided to name the simulations Hyytiälä-diurnal, Hyytiälä-parametrization and CLOUD-parametrization and use this terminology throughout the manuscript now. Parentheses were added.

l.86: This is equation (4), not (3). Please change.

changed.

l.125: Change "For CLOUD at similar..." to "For CLOUD, a similar...".

changed.

l.126: To be consistent, change [HOM_dimer] to [HOM_dim].

adjusted.

l.126: Change "...includes all peaks non-nitrate dimer peaks..." to "...includes all nonnitrate dimer peaks..."

adjusted.

l.136: Change "...cannot necessarily translated..." to "...cannot necessarily be translated...

changed.

l.145: Remove "model" after "(ion-UHMA)".

removed.

l.177/178: The symbols used in Fig. 1b are squares and stars but the figure caption reads "crosses" and "circles". Please revise.

caption adjusted.

l.217: Change "...three characteristics NPF..." to "...three characteristic NPF...".

adjusted.

l.230: Change "significnat" to "significant".

corrected.

l.245: The reference to Figure 4 and the following sentence are incomplete.

sentence completed and reference added.

l.250: The reference to Figure 5 must be corrected.

l.257/258: The referenc to Figure 5c must be corrected.

References to the Figures were corrected.

l.282: Change "udnerestimated" to "underestimated".

changed.

l.282: To be consistent, change [HOM_dimer] to [HOM_dim].

adjusted.

l.291: Change "...rom Eq, (2)..." to "...from Eq. (2)...".

added the missing character.

l.297: Change "...result are closer..." to "results are closer...".

changed.

l.344: Change "...ion-included nucleation..." to "ion-induced nucleation...".

changed.

**Reviewer #2:**

The study by Gonzalez Carracedo et al. reports observations from the boreal forest that show that below 3 nm, the particle growth rate derived from ion observations is lower than that derived from total particle concentration measurements. This work confirms observations from a previous study (Gonser et al., 2014)

and provides an explanation, through the combined analysis of real-atmosphere and laboratory measurements and the use of a process model. Because of its indirect impact on climate, NPF is a key atmospheric process whose study is crucial; the question addressed in this study is of particular interest since the possibility of generalizing observations made from ion measurements (often easier and more direct than neutral particle measurements) to the total particle population is critical. This is an otherwise well written paper, which I therefore recommend for publication in ACP. I suggest however that the comments listed below be addressed before final publication, with, in particular, a clarification in the description of the simulations that are at the heart of this study (Sects. 2.5 and 3.3).

We thank the reviewer for their helpful comments which certainly have helped to improve the clarity of this manuscript.

Comment 1: P1, L29 : I think that a complementary reference to the more recent reviews by Kerminen et al. (2018) or Lee et al. (2019) could shed light on observations made during the last ~20 years, after the review by Kulmala et al. (2004) was published.

We agree with the reviewer and added the more novel references.

Comment 2: P2, L41-42: I would suggest referring to papers that describe the most commonly used methods for GR calculation at this point in the text; in addition to the "maximum" (Hirsikko et al., 2005) and appearance time (Lehtipalo et al., 2014) approaches which are more fully described later in the paper, the work of Dal Maso et al. (2005) could for example be mentioned.

We agree and added the three references at that stage.

Comment 3: P2, L44: In addition to Manninen et al. (2009), I would suggest referring here to the studies by Mirme and Mirme (2013) and Manninen et al. (2016) which are more dedicated to the description of the instrument and its functioning, and which are mentioned elsewhere in the text. Manninen et al. (2016) could also be added at L109.

Again, we agree with the reviewer that these papers are good additions to the text and added the references here.

Comment 4: P2, Sect. 2.1: It would be interesting to indicate / recall (here or later in the results Sect.) how many NPF events were considered in the analysis, for both Hyytiälä and CLOUD datasets.

We agree with the reviewer that giving the statistics provides the reader with more information to put the results into context. We modified the following sections:

"Our field data was collected between March-September 2020 at the SMEAR II station based in the boreal forest in Hyytiälä, Finland (61°51' N, 24°17'E, 181 m a.s.l.), where we recorded 50 NPF events, of which for 18 we could quantify GRs in both the DMA-train and the NAIS from 1.8-8 nm (called "strong" NPF events)."

"(…), using a mixture of sulfuric acid, ammonia, NOx and oxidized organics from alpha-pinene and delta-3-carene ozonolysis as particle precursors (in total 14 experiments)."

Comment 5: P3, Table 1:

- If I am not mistaken, the names given to the simulations (according to the formulation used for J) are only used in Table 1, whereas their use in the text could help when describing successively the different simulations. I think that it should also be made clear that "simple" refers to the prescription of a diurnal profile of J, whereas "lehtipalo" indicates the use of parameterizations;

We thank the reviewer for pointing us towards more clarity here. We decided to rename the models into "diurnal" and "parametrization" and used these terms now consistently throughout the manuscript.

- Why is it indicated "not used" for the HOM dimers concentration while, in line with the formulation of the parameterizations of J (Eqs. 2 and 4), this variable is a priori used in the simulations (e.g.: L251-252)?

The reviewer is correct, that it is the $[HOM_{dimer}]$ which is used as input for the parametrization, however the parametrization in the simulation just takes the total product $[H_2SO_4]^2 [NH_3] [HOM_{dim}]$ as input and the individual terms are not resolved. The HOM species present in the simulation is a nano-Köhler organic compound, which contributes to the growth of the particles and is better related to $[HOM_{tot}]$. We adjusted the text accordingly and write "not used directly" in Table 1.

- Why is there no value for the total HOM concentration in the CLOUD experiments?

We added the corresponding values.

- In the line indicating the product of the precursor concentrations used in Eqs. 2 and 4, [HOM] should be replaced by [HOMdim], right?

Correct. And changed. See above.

- The meaning of $Q_{i.p.}$ and $N_{i.p.}$ should be explained.

Added now to the text in Section 2.5:

"Sub-1.8 nm charged clusters are treated dynamically in the model (i.e. the ion-pair concentration $N_{i,p}$ is the result of a production and loss term, with the latter calculated from ion-ion recombination, dilution and, if applicable, wall losses), with an ion-pair production rate ($Q_{i,p}$) of 3 cm$^{-3}$ s$^{-1}$."

I think that in general, in relation to Sect. 3.3, a clearer description of the simulations would be beneficial to the understanding of the results. However, Sect. 2.5 may come too early in the manuscript for such a detailed description, so one option would be to keep only the general description of the model in Sect. 2.5 and offer a more complete/detailed description of the simulations directly in Sect. 3.3 (and in this case, the last three columns of Table 1 could be removed and made into a table of their own in Sect. 3.3).

We decided to extend the description of the simulations in Sect. 2.5 and Sect. 3.3 (see also reply to referee #1 and later comments) and are confident that the simulation setups are better understandable now.

Comment 6: P3, L84-85: "we also find a parametrization for the ion-induced nucleation fraction based on the ion-pair production rate and the vapor concentrations at which ion-induced nucleation becomes less dominant": the approach used to derive the parameterization is not clear to me, can the authors try to specify / clarify?

We changed the corresponding sentences to:

"We fit the ion-induced nucleation fraction by a function which is limited by the ion-pair production rate (as it gives the maximum rate at which ion-induced nucleation can proceed with every ion seeding a new particle) and which approaches the neutral nucleation rate exponentially around the vapor concentrations where ion-induced nucleation becomes less dominant:

$$J_{1.7}(\text{ion}) \; [cm^{-3}s^{-1}] = c_1 - c_1 \cdot \exp(c_2 \cdot [H_2SO_4]^2[NH_3][HOM_{dim}]) \tag{4}$$

We find $c_1 = 3.4$ cm$^{-3}$ s$^{-1}$ (close to the ion-pair production rate $Q_{i,p}$) and $c_2 = 2 \cdot 10^{-22}$ cm$^{-9}$ pptv$^{-1}$ (free parameter of the fit) using the $J(\text{tot}) = J(\text{ion}) + J(\text{neutral})$ data obtained under galactic cosmic ray conditions (no ion removal in the chamber) from Lehtipalo et al. (2018) for the fit."

**Comment 7:** P4, L86: The equation number should be corrected (4 instead of 3).

Corrected, see above.

**Comment 8:** P4, L94-96: "a good sensitivity towards low particle concentrations in the sub-10 nm range", "the DMA-train can measure also sub-3 nm particle growth with an unprecedented sizing precision": Is there a study in which the performance of the DMA-train has been compared with that of instruments that measure over a comparable size range, including in particular NAIS to also support L169-170 ("where however the total growth rate has generally higher uncertainties due to lower signal when compared to the DMA-train")?

We reference Kangasluoma et al. (2020) to support the argument in L169-170 as it shows nicely that the LOD and overall uncertainty are lower for the DMA-train in the sub-3 nm range.

**Comment 9:** P5, Sect. 2.3:

- Since the mass spectrometry measurements were conducted at a different height than the other measurements (which may therefore raise questions about potentially different conditions with respect to forest canopy), I would suggest referring to Zha et al. (2018), who indicate that during daytime, i.e. the period of interest for NPF, HOM measurements (concentrations and composition distributions) are in fact similar below (i.e. near the ground) and above (i.e. at 35 m height) the canopy;

We agree with the referee here and added the reference and a short sentence. We also point out that we compared some available days of measurement were ground-based CI-API-TOF data were available and mention that total HOM seemed to be similar, but HOMdim was significantly different possibly due to missing transmission corrections:

"Vertical differences for HOMs are typically minor at that measurement site, such that the above canopy measurements can be regarded as representative enough for our near-ground growth estimates (Zha et al., 2018). However, a comparison to few available days of ground-based CI-API-TOF measurements during the campaign revealed lower [$HOM_{dim}$] but similar [$HOM_{tot}$] pointing towards a significantly reduced transmission at large masses."

- I would suggest saying few words on the calibration factor used to convert CI-APi-TOF signals into concentrations in molecules / $cm_3$. At least, was the same coefficient applied to sulfuric acid and HOM signals?

We agree and also extended the discussion on transmission corrections which are missing for Hyytiälä but included at CLOUD. We added:

"Calibration was performed using sulfuric acid (Kürten et al., 2012) and the obtained calibration coefficient ($2.6 \cdot 10^9$) was also used for the concentration measurement of HOM compounds. (…)
For CLOUD, a similar instrument was used and total HOM concentrations were estimated using a similar mass range and included a mass-dependent transmission correction, which was not applied to Hyytiälä data due to missing calibrations."

- L125: "at similar instrument" →   a similar instrument

changed.

- L126: "Lehtiplao" →   Lehtipalo

changed.

Comment 10: P5, L136: "which cannot necessarily translated into a pure condensational growth rate": the wording should be checked.

changed.

Comment 11: P6, L166-167: I take the opportunity of this first illustration of a result that is at the heart of this study to make a more general comment. For a given instrument and calculation method, $GR_{Ions}$ appears to be indeed lower than $GR_{Total}$ below 3 nm. I think it would be interesting to try to quantify (at least roughly) these differences (there is a factor of 2 indicated in the conclusions L338; it would be interesting to mention and discuss this earlier in the manuscript), especially to quickly discuss their magnitude/importance in relation to the uncertainties on the GR estimate related to the different calculation methods and/or the use of different instruments (e.g. Yli-Juuti et al., 2011). In other words, is the estimation error made on $GR_{Total}$ considering $GR_{Ion}$ of the same order, lower, higher, than the differences in the evaluation of $GR_{Total}$ related to the use of different methods / instruments?

We agree with the reviewer that it would be good to shortly also discuss the differences on a quantitative basis. Please see our revised paragraph on the new Figure 1 (response to referee #1) and the updated Fig. 1a where the factor 2 offset should be more visible and we also put it into the context of the overall observed scatter between different GR results. However, Fig. 1a also shows that this variation in the relation between GRion and GRtotal (possibly caused by methodological uncertainties) is still quite significant and hence we refrained from a deeper quantitative discussion than the mentioned factor of 2.

Comment 12: P6, L169: In order to avoid any confusion, I would clearly refer to NAIS: "when using the same instrument (i.e. NAIS)…".

Changed as suggested.

Comment 13: P7, Fig. 1.b: Similar to Fig. 1.a, I would suggest adding the lines delimiting the [1:2, 2:1] range to help in the visual evaluation of the differences between $GR_{Ions}$ and $GR_{Total}$.

The Figure was revised accordingly.

Comment 14: P7, L191: Since the ranges delimited by the boxes / error bars overlap in Fig. 2.a, I would suggest saying "The sub-3 nm ambient ion growth rates are on average clearly lower than the total growth rates…".

Agreed and changed.

Comment 15: P7, L192-193: "However, at larger sizes (3-8 nm), both the laboratory and ambient measurements show no significant differences between the apparent ion and total growth rates (Fig. 2b and 2d)": I do not agree with this statement; the GRs obtained from ion measurements in CLOUD (Fig. 2.d) are on average higher than the total GR, with differences that appear to be of the same order as those found for the ambient measurements in the lower size range (Fig. 2.a).

We agree with the reviewer that the larger size CLOUD ion GRs are slightly above the neutral medians. However as the ion medians aree not above the 75-quantile of the total GRs, we did not identify this as significant (different to the Hyytiälä smaller size range, where the ion medians are below the 25-quantile of the total GRs). We revised the statement:

"However, at larger sizes (3-8 nm), the ambient measurements show no significant differences between the apparent ion and total GRs (Fig. 2b) and the laboratory results even show slightly higher ion than total GRs (Fig. 2d), however the ion medians are not higher then the 75-quantile of the measured total GRs and therefore this effect could be well within potential statistical fluctuations."

Comment 16: P8, Fig. 2:

- Fig. 2.a-d: the meaning of the symbol should be defined explicitly (median, quartiles, range indicated by the error bars, signification of the diamonds);

We changed the caption to explicitly define the Figure content:

"Upper panels (a to d) are box plots of the growth rate distribution during Hyytiälä and CLOUD campaign, showing the median growth rate and 25-75 percentiles of the data as horizontal black line and boxes and the 5-95 percentiles as errorbars and strong outliers as diamonds. Lower panel (e) shows the growth rate measured during both campaigns in relation to the total condensing vapour and their corresponding growth rate according to Eq. (3), with the green squares as the DMA-train data from Hyytiälä and green diamonds the DMA-train data from CLOUD. NAIS growth rate data are displayed as triangles, with the orientation indicating if Hyytiälä or CLOUD data are used."

- Fig. 2.e: In order to ease the reading of the figure, I would suggest keeping the same symbol for Hyytiälä and CLOUD for all instruments, and only use different colours to distinguish between the different instruments.

We think that it clearly enhances the understanding of the Figure, if Hyytiälä and CLOUD datasets can be easily identified. Using the same symbol for both would make it more difficult to understand that there is a clear difference between the two which is the essential point about Fig. 2e.

Comment 17: P10, L230-231: "with the latter becoming more and more significnat at a later stage when also the neutral nucleation pathway dominant": the wording should be checked.

changed to:

"(…), with the latter becoming more and more significant at a later stage when also the neutral nucleation pathway becomes dominant"

Comment 18: P10, L245: "The results are presented in Figure we can clearly observe": the wording should be checked.

changed, see response to referee #1.

Comment 19: P10, Sect. 3.3: Related to Comment 5, I would suggest clarifying the description of the simulations, as well as of some results. Specific points that I would in particular suggest to address are listed below:

We agree with the reviewer and extended our descriptions if the simulations in both Section 2.5 (see reply to referee #1) and Section 3.3.

- Does the "organic concentration" mentioned in L241correspond to $HOM_{tot}$ in Table 1? If so, why are the values given in the text and in Table 1 different ($1\times10^7$ vs $2\times10^7$ cm$_{-3}$)? More broadly, as done for sulphuric acid in L241, the concentrations chosen for the other precursors should at some point be discussed and justified;

We thank the reviewer for finding that discrepancy and corrected it (it is $2\ 10^7$ for both values). And included the orgin of the used concentrations in Section 3.3

- It think it would help to clearly recall at L240-242 that the mentioned species are used for GR calculation in the model (to avoid any confusion with J);

Agreed and adjusted:

"For the condensing vapors contributing to GR, we assumed (…)"

- L249: Eq. (4) was used as well, wasn't it?

Indeed. We added that.

I also strongly suggest to clarify the description of Fig. 5 (expanding / clarifying in particular the sentence that is currently in lines 250-252):

- indicating clearly that the measurements reported on Fig. 5.a are those from the CLOUD experiments which allowed to derive the parametrizations of J used in the simulation;

adjusted:

"The CLOUD results presented in Fig. 5a (Lehtipalo et al., 2018) show that both nucleation pathways (neutral and ion-induced) produce particles across all vapor concentrations, with the neutral nucleation rate scaling with increasing total nucleating vapor."

- specifying how the diurnal profile of the product of the concentration of measured precursors is obtained (average over NPF event days at the station?);

Indeed, this was a sensible point, and we thank the reviewer for spotting it. So far, we used a strong NPF day as the basis for the calculations (apart from the average NH₃ result). We decided to change that to the campaign average over all "strong" NPF days (where GR was obtained for both DMA-train and NAIS over the full size range, which resulted in significantly lower vapor concentrations. At the same time, we reanalyzed our HOM data and could refrain from the usage of the HOM dimer tracer compounds. We therefore deleted these corresponding sections.

Overall, we are now left with a factor of 500 difference for the total vapor concentration at which the transition between ion-induced-dominated and neutral-dominated nucleation occurs. However, as it is shown in the answer to referee #1, also this factor is well within the potential uncertainties of the methods, and we therefore conclude that the results of the manuscript remain fully valid.

- indicating perhaps also clearly that the profile of the fraction of ion-induced nucleation represented in Fig 5.c is obtained by considering the idealized profile of the product of the concentration of precursors shown in the same figure.

We agree with the referee here that this needs to be clarified, especially because the actual profile of the vapors and the speed of the transition between ion-induced and neutral-dominated nucleation determines the significance of the GR effect.

"We therefore idealized the scaled nucleating vapor curve (Fig. 5c) by an analytical Gaussian expression for computational simplicity and used it as input for the nucleation rate calculation with the parametrizations given in Eq. (2) and Eq. (4)."

Comment 20: P12, L282: udnerestimated → underestimated

changed.

Comment 21: P13, Fig. 6: the diurnal profile of sulfuric acid concentration shown in Figure 6.d appears to be different from that shown in Figure 4.d, and unless I am mistaken it is not indicated in the text that the sulfuric acid profile is different in the two simulations. Can the authors provide an explanation?

The profiles are different as for Fig. 6, the sulfuirc acid time-dependence is chosen to follow the same time-dependence as it is observed for the total set of vapors in Fig. 5c. To avoid confusion, we clarified this in the text:

"We kept the vapors concentrations for the vapors contributing to growth ($H_2SO_4$ and nano-Köhler organics similar to $HOM_{tot}$) similar as for the Hyytiälä-diurnal simulation (with $H_2SO_4$ also changed to a Gaussian profile better resembling the actual diurnal pattern)"

Comment 22: P14, Conclusions : This study is dedicated to the boreal forest but do we have any idea / can we anticipate the observations that could be made in other environments and in particular at high altitude, where the role of ions in nucleation is a priori more marked than at low altitude (Manninen et al., 2010; Sellegri et al., 2019)? Can we assume that we have less error in $GR_{total}$ based on ion observations there, or is it difficult to anticipate without detailed knowledge of the precursors involved?

Our results show that the difference between GRion and GRtotal is induced by the time for the transition between an ion-induced-dominated and neutral-dominated nucleation regime. If ion-induced nucleation remains the dominant mechanism throughout the day (as the reviewer suggests for high altitudes), we do not expect any difference between GRion and GRtotal. In the same sense, one could argue that if neutral nucleation is always the dominant pathway (i.e. in polluted urban environments where nucleation proceeds via H2SO4-amine clusters), the period where ion-induced nucleation dominates in the early morning might be very short and hence also no bigger differences between GRion and GRtotal are expected. However, as the reviewer remarks already, these speculations require detailed knowledge on the nucleation mechanism. We thus added two sentences on these thoughts to our final conclusions, to not speculate too much on these topics:

"However, the effect of reduced ion GR compared to total particle GR depends on the actual nucleation mechanism, the abundance of ions and ultimately on the transition time between an ion-induced and neutral dominated nucleation regime and hence it is difficult to transfer our findings to other environments. In settings where ion-induced nucleation always dominates (e.g. at remote or high-altitude sites) or where the transition is very short due to a strong neutral nucleation pathway (e.g. polluted settings with strong H2SO4-amine clustering), we expect to see less differences between ion and total GR."